# Evolutionary Implications of the microRNA- and piRNA Complement of *Lepidodermella squamata* (Gastrotricha)

**DOI:** 10.3390/ncrna5010019

**Published:** 2019-02-22

**Authors:** Bastian Fromm, Juan Pablo Tosar, Felipe Aguilera, Marc R. Friedländer, Lutz Bachmann, Andreas Hejnol

**Affiliations:** 1Science for Life Laboratory, Department of Molecular Biosciences, The Wenner-Gren Institute, Stockholm University, S-10691 Stockholm, Sweden; marc.friedlander@scilifelab.se; 2Functional Genomics Unit, Institut Pasteur de Montevideo, Montevideo 11400, Uruguay; jptosar@pasteur.edu.uy; 3Nuclear Research Center, Faculty of Science, Universidad de la República, Montevideo 11400, Uruguay; 4Departamento de Bioquímica y Biología Molecular, Facultad de Ciencias Biológicas, Universidad de Concepción, Casilla 160_C, Concepción 3349001, Chile; faguilera@udec.cl; 5Sars International Centre for Marine Molecular Biology, University of Bergen, 5006 Bergen, Norway; andreas.hejnol@uib.no; 6Research group Frontiers in Evolutionary Zoology, Natural History Museum, University of Oslo, 0318 Oslo, Norway; lutz.bachmann@nhm.uio.no

**Keywords:** microRNAs, piRNAs, RNAi protein machinery, Gastrotricha, *Lepidodermella squamata*, Platyhelminthes

## Abstract

Gastrotrichs—’hairy bellies’—are microscopic free-living animals inhabiting marine and freshwater habitats. Based on morphological and early molecular analyses, gastrotrichs were placed close to nematodes, but recent phylogenomic analyses have suggested their close relationship to flatworms (Platyhelminthes) within Spiralia. Small non-coding RNA data on e.g., microRNAs (miRNAs) and PIWI-interacting RNAs (piRNA) may help to resolve this long-standing question. MiRNAs are short post-transcriptional gene regulators that together with piRNAs play key roles in development. In a ‘multi-omics’ approach we here used small-RNA sequencing, available transcriptome and genomic data to unravel the miRNA- and piRNA complements along with the RNAi (RNA interference) protein machinery of *Lepidodermella squamata* (Gastrotricha, Chaetonotida). We identified 52 miRNA genes representing 35 highly conserved miRNA families specific to Eumetazoa, Bilateria, Protostomia, and Spiralia, respectively, with overall high similarities to platyhelminth miRNA complements. In addition, we found four large piRNA clusters that also resemble flatworm piRNAs but not those earlier described for nematodes. Congruently, transcriptomic annotation revealed that the *Lepidodermella* protein machinery is highly similar to flatworms, too. Taken together, miRNA, piRNA, and protein data support a close relationship of gastrotrichs and flatworms.

## 1. Introduction

The Gastrotricha is an animal taxon with hitherto around 700 described species that are divided into the two main clades Macrodasyida and Chaetonotida [1,2,3]. Gastrotrichs are microscopic animals that occur abundantly in marine and freshwater interstitial environments [1,4]. They show a direct development, and are bilaterally symmetric, with an acoelomate body, and worm-like body plan [5]. Most gastrotrich species have a commissural brain [6,7,8,9] and within the clade there is a large variety of reproductive strategies [5,10]. Interestingly, most gastrotrichs lack an ectodermal hindgut, and a specialized respiratory system has not been described [5]. Gastrotrichs share complex morphological characteristics, including a radial myoepithelial pharynx with terminal mouth, epidermal monociliation, and protonephridial structures, with several other lineages within Bilateria. This has made their phylogenetic position a matter of debate. [11]. Initially, gastrotrichs were considered to be close to rotifers (“Trochelminthes”) [12]. Later, classic systematists placed gastrotrichs either as the sister group of Introverta, together forming Cycloneuralia (i.e., Kinorhyncha + Loricifera + Priapulida + Nematoda + Nematomorpha) [13,14], or in the Neotrochozoa group that comprises Gnathostomulida + Gastrotricha [15,16]. When nucleotide sequence data became available, gastrotrichs were placed within Spiralia with uncertain relatedness, within the debated group of ‘Platyzoa’ comprised of Platyhelminthes, Gnathifera, and Gastrotricha [17,18,19]. Recent phylogenomic studies suggest a sister group relationship between Gastrotricha and Platyhelminthes (Rouphozoa) [20,21]. However, this Gastrotricha-Platyhelminthes clade is still controversial because there is no morphological apomorphy supporting the monophyly of the clade [22]. Even though gastrotrichs and most flatworms ingest food with a simple pharynx [6,23,24], this common alimentary strategy is not necessarily an autapomorphy of Gastrotricha and Platyhelminthes. Thus, additional lines of evidence are required to further test the proposed close phylogenetic relationship of gastrotrichs and flatworms.

MicroRNAs (miRNAs) are a unique class of short non-coding RNAs with major roles as post-transcriptional gene regulators in animals and plants [25]. In contrast to short interfering RNAs (siRNAs), responsible for RNA interference (RNAi), and PIWI-interacting RNAs (piRNAs), responsible for genome integrity [25,26], miRNAs are deeply conserved in metazoans. Their continuous addition to genomes during evolution has often been correlated with increasing morphological complexity and number of cell-types in a given organism [27,28,29,30]. Accordingly, absence/presence of miRNAs has also been successfully applied as phylogenetic and taxonomic arguments [31,32,33,34], and were also used to ascertain clade specificity of highly derived organisms such as Myzostomida [35]. However, because miRNA annotation is a challenging task, many published miRNA complements have considerable issues with respect to quality and completeness [36,37]. For example, missing miRNA families have frequently been misinterpreted as secondary losses, and the conservation and the utility of miRNAs as phylogenetic markers has been questioned [38]. However, losses of miRNA families have rarely been observed in well-annotated complements, and the few confirmed cases were correlated to high degrees of genomic and phenotypic reductions, for instance in parasitic species [39,40,41]. Currently, information on the presence of the RNAi pathway and small RNAs in gastrotrichs is lacking, but several studies have reported on miRNAs, piRNA, and the RNAi protein machinery in flatworms [40,42,43,44,45,46]. Closer scrutiny of these studies revealed a dynamic evolution of RNAi proteins [44,45] and small RNAs, with the substantial loss of conserved and gain of novel miRNAs [40], and the loss of piRNAs in obligate parasitic flatworms and nematodes [43,44,47]. In free-living nematodes such as *Caenorhabditis elegans*, however, piRNAs have been detected and described in detail [48]. With respect to miRNAs there is substantial different between flatworms and nematodes. In addition to the miRNA families found in all protostome species, flatworms share two miRNA families MIR-1989 and MIR-1992 with spiralian taxa (we are following the nomenclature proposed by Fromm et al. [36]). Nematodes are not part of Spiralia and, thus, do not share MIR-1989 and MIR-1992 but show instead additional four miRNA families specific for Ecdysozoa (MIR-305) and Nematoda (MIR-54, MIR-86, MIR-791), respectively. Thus at least six miRNA families differ between flatworms and nematodes. It is also noteworthy that, although free-living flatworms and nematodes have both piRNAs, the actual mature piRNAs and the protein machinery in nematodes are unique, and so far do not resemble what has been described for any other animal taxon. Thus, piRNAs of free-living flatworms and nematodes are clearly distinguishable. Using small-RNA sequencing and available transcriptome data, we here describe the first small RNA complement (miRNA and piRNA) as well as the corresponding protein machinery for a gastrotrich, namely the freshwater species *L. squamata*. The analyses revealed that the genome of *L. squamata* encodes an extensive small RNA repertoire of *bona fide* miRNAs and piRNAs, as well as the full miRNA & piRNA biogenesis and RNAi protein machinery. The comparison of miRNAs, but also of piRNAs and RNAi protein machinery, to those of flatworms and nematodes together supports a close relationship of gastrotrichs to flatworms and not to nematodes. 

## 2. Results

### 2.1. The microRNA Complement of Lepidodermella squamata

Small-RNA-sequencing yielded 22.4 millions raw and 217,000 unique high quality and trimmed miRNA reads (SRA Accession number: PRJNA511377). These reads were matched against 194.5 millions of high-quality genomic reads provided by the *L. squamata* genome sequencing project (Aguilera et al. in prep) using the miRCandRef pipeline [40]. In total 20,375 miRNA-candidate contigs were assembled from the genomic reads with perfect matches to small RNA reads, all representing potential miRNA loci. Subsequently, this miRNA-candidate reference assembly was screened with an adapted version of the MirMiner pipeline [31] for miRNA loci with longer hairpins (Fromm et al. in prep). Forty-six *L. squamata* miRNAs that belonged to 35 conserved families and 6 miRNAs with yet undescribed seed sequences were identified, all of them fulfilling the latest criteria for miRNA annotation [36] (see Appendix A). Given the undisputed protostome affinity of *L. squamata*, a set of conserved miRNA families was expected [31,32]. Indeed, we found the Eumetazoa-specific miRNA family (MIR-10) (Figure 1a), 25 of the 31 Bilateria-specific families and 8 of the 11 Protostome-specific miRNA families. We did not find any representatives of the four Nematoda-specific miRNA families, but there was a representative of the Spiralia-specific family MIR-1992 (Figure 1a,b), which is in line with the hypothesis of a close relationship of gastrotrichs to flatworms [20,21]. Interestingly, we found seven miRNAs of conserved families with particularly long hairpins (Let-7, Mir-750, Mir-22, Mir-31, Mir-1992, Mir-1) that exceeded commonly observed hairpin-sizes of 60–90 nucleotides (nt) with miRNA precursor lengths up to 386 nucleotides (Mir-750) (Figure 1b,c). Such long precursors are so far only described in flatworms [40,49,50,51].

The miRNA complement of *L. squamata* was submitted to MirGeneDB and will be available in one of the next releases. 

### 2.2. Lepidodermella squamata Has a Conventional Animal piRNA Biogenesis Mechanism

Four putative piRNA clusters composed of 1377 unique reads and with a total length of 18,789 bp were identified (Appendix A). These piRNAs were identified using stringent criteria for piRNA cluster definition to minimize false positives. 

Clusters #1–#3 include piRNAs deriving from a single DNA strand, and the vast majority of the sequences had a length of 30–32 nt (Figure 2b). In contrast, cluster #4 comprised reads mapping to both strands (minus > plus). Minus strand-derived reads in cluster #4 had a similar length distribution as those in the unistrand clusters #1–#3. However, the length distribution of plus strand-derived reads was clearly bimodal, ranging from 23–26 nt and 30–32 nt, respectively (Figure 2b,c). Overlap with transposable elements (TEs) was partial in some of the clusters, and cluster #4 was remarkably devoid of identifiable TEs (Appendix A).

Main strand-derived sequences in the four clusters were strongly biased towards 5′ uridine (1T in our data), while 5′ ends in the antisense (plus strand) sequences in cluster #4 were virtually random (1T = 25%). Conversely, these sequences showed 10A bias (Figure 2d). The presence of sense sequences, with strong 1T bias and antisense sequences with evident 10A bias, strongly suggested the generation of initiator and responder piRNAs by a conventional ping-pong amplification cycle. This was confirmed by the observation that sense and antisense sequences in cluster #4 overlapped by exactly 10 nucleotides (Figure 2e). 

A more detailed analysis of the genomic distribution of mapped reads in cluster #4 was also consistent with the generation of trailing piRNAs by a piRNA-independent endonuclease acting on large piRNA precursor transcripts (reviewed in [52]) (Figure 2f). Such piRNAs are phased by a length which comprises the footprint of a putative PIWI protein plus a few nucleotides to the next available uridine in the 5′-3′ direction. As observed in most minus strand-derived sequences, these putative trailing piRNAs were 30–32 nt in length. In contrast, responder piRNAs generated from the opposite strand (or from antisense transcripts present in *trans*) are either of this size or 24–25 nt long. This has been explained by the co-existence of at least two different PIWI-clade proteins (see below) acting on piRNAs derived from this cluster, with the proteins being responsible for the shorter piRNAs acting exclusively on transcripts that are antisense to the initial piRNA precursor. In summary, a conventional piRNA pathway involving all the characteristic features of the PIWI processing machinery was found in *L. squamata* that is distinct from the situation in most nematodes [48]. 

### 2.3. RNAi Protein Machinery of Lepidodermella squamata and Gastrotricha

The small non-coding RNA pathways are best described for *C. elegans* [53]. A systematic survey for protein repertoires directly involved in RNA interference pathways was performed using a curated set of 65 RNAi proteins as queries (see Appendix A). We found putative orthologs of most *C. elegans* RNAi protein complexes, except for Secondary Argonautes (SAGOs) (Figure 3a). 

Using a phylogenetic framework the distribution of RNAi protein repertoires within Spiralia was targeted. While the core of the RNAi pathways was generally conserved, several specific factors were absent. They may diverged to a degree that we could not identify the respective orthologs any longer in Gastrotricha (Figure 3a). Focusing on six gastrotrich species including *L. squamata*, we found that Gastrotricha possessed putative orthologs of at least one gene from each of the respective nematode RNAi complexes (Figure 3a), corroborating that gastrotrichs have the complete RNAi protein machinery involved in miRNA biogenesis. As expected, we found multiple hits of Argonaute proteins (*ALG-1* and *ALG-2*) for all gastrotrich species, except for *Mesodasys laticaudatus* (Figure 3a and Appendix A). We also confirmed the complete protein sets of the Microprocessor (*DRSH-1* and *PASH-1*) and Dicer complexes (*DCR-1* and *DRH-1*), except for *RDE-4* gene that is conserved only in some nematode species [58,59] and also absent in flatworms [45] (Figure 3a). Possible duplicates of *DCR-1* were found in the transcriptomes of *L. squamata* (Chaetonotida) and *Macrodasys* sp. (Macrodasyida) (Appendix A). Besides these core RNAi proteins, we found that *L. squamata* encoded the full canonical protein repertoire of the animal RNAi pathway, while in other gastrotrich species we could not find the complete miRNA protein sets (Appendix A). However, transcriptomes can only be used to identify RNAi genes, but not to conclude that a particular gene is missing from the genome.

A screen for the presence of functional domains of some core proteins revealed that *L. squamata DCR-1* proteins had a helicase conserved C-terminal domain, a PAZ domain, a dicer dimerization domain, and two ribonuclease III domains. (Figure 3b). The predicted *L. squamata DRSH-1* and *PASH-1* proteins had a conserved organization, with two ribonuclease III domains and a double-stranded RNA binding motif in Drosha and a double-stranded RNA binding motif in Pasha (Figure 3b). The domain organization of these proteins resembled largely to what has been described for similar proteins in flatworms but only to a minor degree to those of nematodes [45]. The organization of functional domains in *XPO-1*, a transmembrane channel protein that transports pre-miRNAs into the cytoplasm in *C. elegans* [60], also more conserved between gastrotrichs and platyhelminths rather than between gastrotrichs and nematodes (Figure 3b). 

An analysis of PIWI genes targeted the proteins involved in the organization of the four piRNA clusters found in *L. squamata*. Systematic Blast searches of predicted proteins from animal genomes and gastrotrich transcriptomes identified 180 sequences corresponding to PIWI-like genes across metazoan species and confirmed the presence of seven PIWI-like proteins in Gastrotricha with PAZ and Piwi domains (Appendix A). Alignments of these functional domains across *L. squamata* and representatives of Platyhelminthes and Ecdysozoa revealed several conserved residues, that provided further support for considering these sequences PIWI-like proteins (Appendix A).

Phylogenetic analysis (maximum likelihood) of PIWI-like proteins revealed good support for taxon-specific clades, but with only poor support for early diverging branches (Figure 4). Specifically, in *L. squamata*, we found that five PIWI-like proteins were grouped into three phylogenetic clades (Figure 4). Three PIWI-like paralogs (i.e., *Lsqu.Gene.78021, Lsqu.Gene.95003*, and *Lsqu.Gene.176266*) were grouped with *Drosophila melanogaster Piwi* and *Aubergine*, and *C. elegans PRG-1* and *PRG-2*, but with low statistical support. Similarly, the *Lsqu.Gene.203941*, along with a *Diuronotus aspetos* PIWI-like protein, was clustered with *D. melanogaster AGO3*, but also with low bootstrap support (Figure 4). Interestingly, the *Lsqu.Gene.20027* protein that grouped with strong support (i.e., 100% bootstrap) with proteins from platyhelminth species (i.e., *Schmidtea mediterranea, Echinococcus multilocularis*, and *Hymenolepis microstoma*) (Figure 4). While we did not find *L. squamata* representatives of the *Piwil1* and *Piwil2* clades, we found one PIWI-like protein from *Megadasys* sp., that is part of the *Piwil2* clade (Figure 4). Finally, we found two clade-specific groups comprising PIWI-like proteins from *S. mediterranea* and *Macrostomum lignano*. 

## 3. Discussion

Gastrotrichs are an abundant group of organisms in marine habitats with important roles in aquatic ecosystems [1,4]. However, only little is known about their molecular biology [5]. In order to obtain a better molecular understanding of this, and to contribute to the discussion on their phylogenetic relatedness to other protostome species, we have studied the miRNA and piRNA gene-regulatory pathways in the chaetonotid gastrotrich *Lepidodermella squamata* using a ‘multi-omics’ approach. 

The miRNA complement of *L. squamata* consisted of highly conserved and some novel miRNAs. There was an almost complete set of the expected miRNA families and the spiralian-specific MIR-1992, but no Ecdysozoa- or Nematoda-specific miRNAs were detected. Some of the few losses observed in *L. squamata* were shared with all other flatworms studied so far [40,49,61,62]. Interestingly, some of the gastrotrich miRNAs showed extended precursor lengths of up to 386 nucleotides, which, to this date, has only been described in flatworms [40,51,63]. Taken together, we found a near complete set of miRNAs that by the presence and absence of particular miRNA families, their high sequence similarities and the length of some precursors support a close relationship of gastrotrichs and flatworms. 

The conservation of individual piRNA sequences is extremely low [64,65,66] and thus their potential for evolutionary or phylogenetic analyses is thought to be limited. However, we showed that similarities in piRNA biogenesis patterns can be used to infer relatedness of animal groups such as the gastrotrichs to flatworms or nematodes. While both free-living flatworms and free-living nematodes possess piRNAs, many parasitic lineages have lost them altogether [48] or, in the case of nematodes, have retained a highly derived pathway, producing piRNAs that are clearly distinct from most other animals studied thus far [52]. For instance, piRNAs in *C. elegans* are not produced from long primary transcripts derived from genomic regions known as piRNA clusters and have a size of 21 nucleotides instead of the conventional 24–32 nt lengths typical of piRNAs in other animals [67]. 

Although, the four identified piRNA clusters do not constitute the full piRNA complement of *L. squamata*, they already allowed for some insights into the piRNA processing machinery in Gastrotricha. Three piRNA clusters with piRNAs mapping to only one DNA strand and a fourth cluster including piRNAs from both DNA strands were identified. The piRNAs of the latter cluster showed processing patterns consistent with the so-called ping-pong amplification mechanism [52], and the majority of the piRNAs were derived from the minus strand and showed 1T bias. In contrast, plus strand-derived piRNAs, which are antisense to the putative precursor transcript, showed strong bias towards adenine at the 10th position and overlapped minus strand-derived piRNAs by exactly ten nucleotides. It is known that the 10A bias is caused by the intrinsic affinity of PIWI-clade proteins for targets bearing adenine at position target-1 or t1 (i.e., in front of the piRNA 5′ end). When PIWI cleaves their targets in a piRNA-directed manner, it does so at position 10 of the target counting from the piRNA 5′ end. As a consequence, adenine at position t1 becomes adenine at position guide-10 or g10 [68]. piRNAs generated in this way were initially termed as secondary piRNAs, although the term ‘responder piRNA’ has been recommended [52]. In *Drosophila melanogaster* (an ecdysozoan), the ping-pong cycle involves two of the three PIWI-clade proteins present in the genome. Aubergine (*Aub*) binds piRNAs produced from piRNA precursors while Argonaute 3 (*AGO3*) associate with complementary transcripts, usually corresponding to TEs. Aub-bound sequences tend to start with uridine, while AGO3-associated piRNAs show 10A, but no 5′ bias [69]. A similar heterotypic ping-pong amplification cycle was also recently described in mollusks, where two different PIWI proteins bind sense and antisense transcripts, respectively [70]. These two proteins bind piRNAs of 29–30 nt (with 1U bias) and 24–25 nt (with 10A bias), respectively. Interestingly, homotypic ping-pong amplification was also described in the oyster *Crassostrea gigas*, involving a single PIWI-clade protein acting on both sense and antisense transcripts. As a consequence of concomitant homotypic and heterotypic ping-pong amplification in the gonads of *C. gigas*, 1U-biased piRNAs showed a size distribution of 29–30 nt, while partially complementary, 10A-biased piRNAs were either 24–25 (heterotypic) or 29–30 (homotypic) nt long. 

The piRNAs of cluster #4 piRNAs in *L. squamata* (Figure 2a–e) meet this pattern, with the only difference that the 1U-biased piRNAs derived from the putative piRNA precursor transcript were slightly longer in Gastrotricha (30–32 nt vs 29–30 nt, Figure 2a,b). Mammalian PIWI proteins are also associated with piRNAs of slightly different lengths. For instance, mouse piRNAs bound to MILI or MIWI2 show mean lengths of 26 and 28 nt, respectively [71]. 1U-biased piRNAs in *L. squamata* tend to be slightly longer than in the majority of animals analyzed to date [70,71,72]. They are very similar to what was previously reported in the planarian *S. mediterranea* [42], where bands of 31–32 nt were identified by both sequencing and Northern blot analyses. However, an exhaustive analysis of piRNA lengths across metazoan species is still lacking.

Although we cannot exclude the existence of 21U piRNAs (as not all 21 nt reads were assigned to other classes), the presence of canonical piRNAs and their genomic distribution in piRNA clusters suggest that piRNA processing patterns in *L. squamata* is different than in nematodes, but, instead, very similar to what was recently reported for mollusks [70] (a spiralian) and comparable in size to the piRNA complement in flatworms [42,45]. Accordingly, the piRNA data indicate that Gastrotricha (*L. squamata*) are more closely related to Platyhelminthes within Spiralia rather than to Nematoda within Ecdysozoa. 

A strict homology approach to the protein repertoires of the RNAi and PIWI protein machinery across protostomes (i.e., spiralians and ecdysozoans) corroborated that nematodes (including *C. elegans*) have a more derived RNAi pathway [52] compared with other spiralians. Among spiralians, we found that RNAi proteins from gastrotrichs (including *L. squamata*) and flatworms are very similar in terms of RNAi protein phylogenetic distribution and domain architecture, supporting their closer relationship and the Rouphozoa phylogenetic clade [20,21]. On the other hand, our analyses on PIWI protein repertoires in *L. squamata* revealed all necessary protein machinery for piRNA biogenesis [25]. In addition, the presence of several PIWI-like proteins, some of them with similarity to *D. melanogaster Aub* and *AGO3*, suggest that piRNAs might be controlled by a system similar what has been observed in Arthropoda. Although further experimental evidence is needed to support this hypothesis, we demonstrated that *L. squamata* PIWI repertoire is not similar to nematodes studied so far. Taken together, all data inferences covering reciprocal BLAST, functional domain annotation, and phylogenetic analysis, strongly support the close phylogenetic relationship between Gastrotricha and Platyhelminthes, which is congruent with miRNA and piRNA results.

In conclusion, we show that the freshwater gastrotrich *Lepidodermella squamata* (Chaetonotida) expresses a comprehensive set of bona fide miRNAs and piRNAs and the corresponding biogenesis and RNAi machinery. Furthermore, we demonstrate that the occurrence, structure and sequence of miRNA genes and families is useful to assess the phylogenetic position of gastrotrichs and strongly supports close relatedness to flatworms and not to nematodes. Congruently, by comparing the length and structure of mature and precursor piRNAs, we show that the presence of bona fide piRNAs and ‘responder piRNA’ in particular, clearly supports a closer relationship to flatworms, as well. This is consistently followed by the presence/absence distribution and domain architecture of RNAi proteins supporting this Rouphozoa clade, too. 

With miRNA and piRNA complements for only one species we certainly cannot make statements about the monophyly of gastrotrichs [20]. Nevertheless, these first small non-coding RNA complements of *L. squamata* add important information for an understanding of the evolution of non-coding RNAs across the metazoan tree of life. We showed that not only the highly conserved miRNAs, but also the piRNAs and the corresponding protein machinery can add valuable phylogenetic information in support of a close relationship of *L. squamata* to flatworms. 

Our knowledge on the evolution of ncRNAs is broad but not very detailed because the annotation of bona fide ncRNA complements, especially for microscopic organisms, remains a challenge as it requires material from often very sparse samples, expert knowledge, and manual curation. Surprisingly, even major endeavors such as the International Helminth Genomes Consortium have neglected ncRNA analyses despite the obvious availability of high quality genome assemblies [73]. In the absence of such assemblies, our pipeline miRCandRef can be used for miRNA and piRNA annotations that can be complemented by information of key RNAi and biogenesis proteins through readily available transcriptome assemblies for all major metazoan groups toward a better understanding on the evolution of coding and non-coding elements. 

## 4. Materials and Methods

### 4.1. Lepidodermella squamata Culture 

500 adult specimens of *L. squamata* (Dujardin 1841) [74] were isolated from a purchased culture from Connecticut Valley Biological Supply Co., Inc., Southampton, MA, USA. Animals were maintained on wheat grains that contained a variety of bacteria, protozoa, and fungi, but no other metazoans. 

### 4.2. small-RNA Sequencing

A total RNA sample enriched for small RNA was extracted from 500 pooled individuals using the ZR RNA MicroPrep kit (Zymo Research, Irvine, CA, USA). miRNA library construction service from total RNA using Illumina TrueSeq Small RNA kit and sequencing on a NextSeq 500 Mid instrument was outsourced to StarSEQ GmbH, Mainz, Germany. 

### 4.3. small-RNA Bioinformatics 

miRTrace [75] was used to process the raw small RNA-Seq reads. The algorithm miRCandRef (https://hyperbrowser.uio.no/mircandref/) [40] was used for assembling high-quality genomic reads into relatively short genomic contigs (crystal-contigs) that constitute candidate small-RNA loci. Briefly, the pipeline delivers a multifasta file of assembled and clustered crystal-contigs that can be used as a reference for piRNA and miRNA prediction with software like proTRAC [76] and MirMiner [31] or miRDeep2 [77] for miRNAs. To distinguish between missing and not expressed miRNAs, the contigs were additionally screened for absent miRNA families by BLAT [78] as well as for spiralian miRNA families from MirGeneDB 2.0 [36,79].

Putative piRNAs were identified by mapping small RNA-Seq reads to the *L. squamata* genome assembly (Aguilera et al.; in prep) with sRNAmapper [80]. By prioritizing specificity over sensitivity to avoid confounding between piRNAs and other ncRNAs or their fragments [81], we removed reads mapping to either miRNA precursors or predicted tRNA and rRNA reference (with a mismatch allowance of up to three nucleotides) obtained with tRNAScan-SE [82] and RNAmmer [83], respectively. Sequences mapped to miRNA genes were also removed. proTRAC [76] was then used to identify non-annotated sequences showing 1T and/or 10A bias (>75% of the sequences), bias towards a piRNA-like size-range (> 75% of the sequences being 24–32 nt) and forming clusters of at least 500 bp with a minimum of 20 hits per cluster and an even distribution in sequence abundance (so that the top 10% of reads account for less than 75% of the reads in the cluster). 

### 4.4. Identification and Domain Architecture of RNAi Proteins

We obtained available genome-derived proteomes from a wide range of metazoans focusing on spiralian species (Appendix A). For *L. squamata* and other gastrotrich species that predicted proteomes are not available, raw reads were downloaded from NCBI SRA database (Appendix A), quality trimmed using Trimmomatic v0.36 [84] (default settings) and assembled using Trinity v2.1.1 [85] (with the -normalized_reads option set for *L. squamata* and *D. aspetos*, otherwise default settings), Trans-ABySS v1.5.3 [86] (default settings), and Bridger v2014-12-01 [87] (default settings). Transcriptome assemblies were combined and redundant transcripts were clustered using CD-HIT v4.6 and TransDecoder v3.0.1 following the strategy described by [88], and the longest ORFs were selected for further analyses.

To search for the presence of RNAi proteins across metazoan species, we downloaded the *C. elegans* proteome from WormBase (www.wormbase.org; release WS267) and a total of 65 proteins involved in small non-coding RNA pathways [45] was used as queries for BLASTp searching with an E-value threshold of 10^−5^ [89]. Positive hits were then verified by reciprocal Blast searches against the *C. elegans* proteome (using the same E-value threshold), and only proteins ranked as top hit by both approaches were retained. The results for flatworms were compared with recent findings by Fontenla et al. [45] and all hits were further verified by Blast search against GenBank nr database.

Proteins retrieved as Dicer, Drosha, Pasha, and Exportin from *L. squamata* and some representatives from Annelida, Platyhelminthes, Rotifera, and Nematoda were thereafter classified and annotated by using Pfam database (Pfam-A version 32.0) [57]. Sequence comparisons were performed by visual inspection and all domain architecture figures were created using Adobe Illustrator CC 2015.

### 4.5. Identification and Phylogenetic Analysis of PIWI Proteins

To reconstruct the phylogenetic relationship of gastrotrich PIWI-like proteins, we searched for PIWI genes in different protostome genomes and gastrotrich transcriptomes (Appendix A). To this end, we looked for annotated PIWI genes of the nematode, *C. elegans* (D2030.6 and C01G5.2a, release WS244); Pacific oyster, *Crassostrea gigas* (EKC35279 and EKC29295); fly, *D. melanogaster* (AGL81535, AGA18946, and NP_001163498); and human, *Homo sapiens* (NP_004755, NP_060538, NP_001008496, and NP_689644) by using Blastp with a cut-off of 10^−5^. Those proteins with positive hits were selected and checked for presence of PAZ and Piwi domains using Pfam database [57]. 

The predicted and annotated PIWI protein sequences from Gastrotricha, Platyhelminthes, Phoronida, Annelida, Nemertea, Orthonectida, together with PIWI paralogs of nematode, fly, and human, were aligned using MAFFT v7.407 [90] with the E-INS-i algorithm and the BLOSUM45 scoring matrix. Non-conserved regions were automatically removed with TrimAl v1.4 [91], and maximum likelihood (ML) analysis was run using RAxML v8.2.4 [92] with PROTGAMMAAUTO and autoMRE options. Support values were generated by bootstrap with 300 replicates and phylogenetic tree was visualized and edited using the Interactive Tree Of Life web-based tool [93]. 

## Figures and Tables

**Figure 1 ncrna-05-00019-f001:**
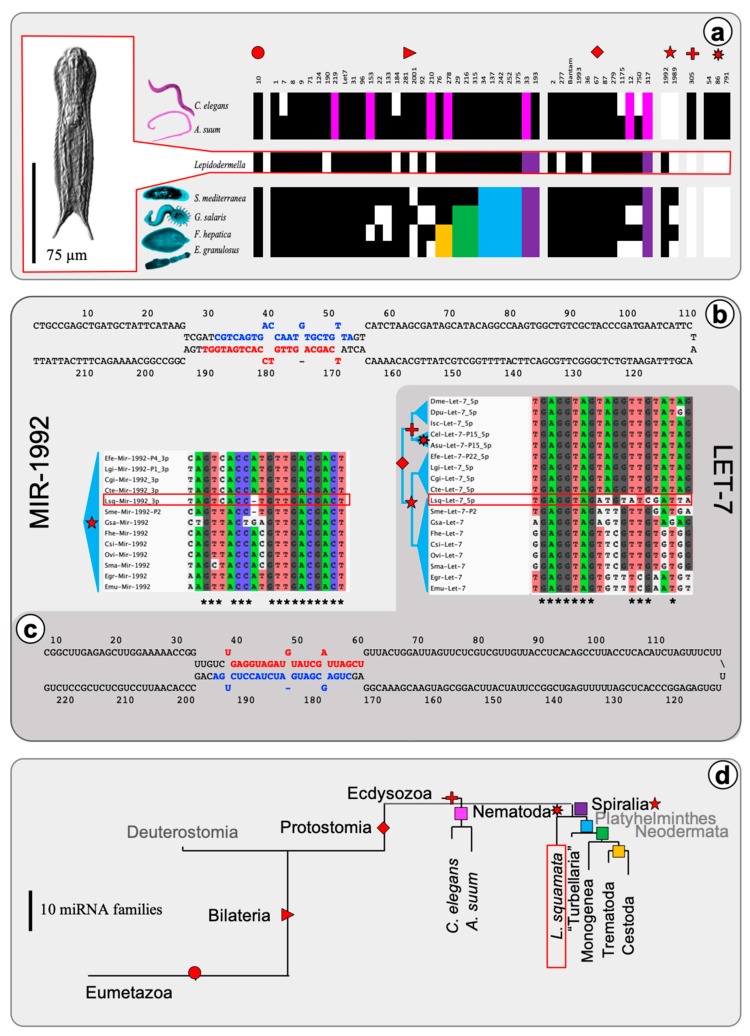
The miRNA complement of *L. squamata* shows a strong affinity to flatworms. (**a**) miRNA distribution in nematodes (top), *L. squamata* (middle and red; depicted in red box to the right) and flatworms (bottom) show pattern of taxa-specific gains and losses of conserved miRNA families for each group. Presence/absence pattern of taxa-specific groups of conserved miRNAs are depicted in red icons (Eumetazoa, Bilateria, Protostomia, and Spiralia); compare to (d), black squares represent the presence of a miRNA family. Losses are depicted as white fields or highlighted in colors when interpreted as shared losses (pink: Nematoda, purple: Spiralia, blue: Platyhelminthes, green: Neodermata, yellow: Trematoda+Cestoda). (**b**) (top) Structure of extended precursor of Lsq-Mir-1992, a spiralian-specific miRNA family, including mature (red) and star (blue) annotation and (bottom left) sequence alignment with all known members in Spiralia (3′-5′ direction). (**c**) (bottom) Structure of extended precursor of Lsq-Let-7, a bilaterian wide distributed miRNA family, including mature (red) and star (blue) annotation and (top right) sequence alignment with selected invertebrate members. (**d**) Gain and loss of miRNA families in Eumetazoa, Bilateria, Protostomia, Nematoda, Platyhelminthes, and *L. squamata* branch lengths corresponds to number of gains/losses.

**Figure 2 ncrna-05-00019-f002:**
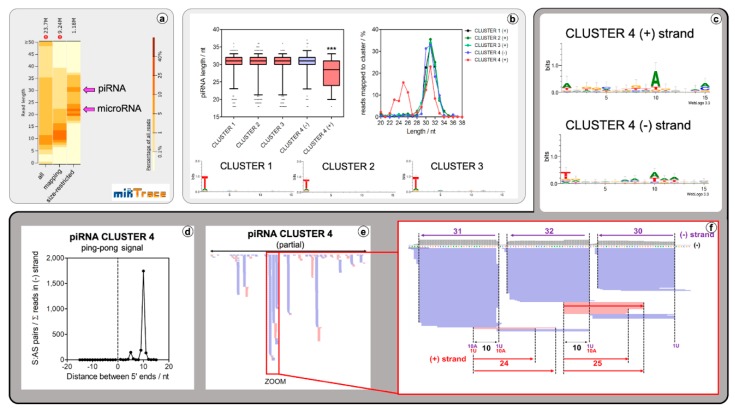
piRNA discovery in *L. squamata.* (**a**) miRTrace read-length distribution of all, genome mapped and genome mapped plus size-selected reads (20–37 nt) show that distinct peaks for bona fide miRNA and genuine piRNAs are hidden by a body of non-small-RNA fragments. (**b**) Top left: size distribution of piRNAs assigned to each cluster, discriminating between piRNAs derived from the plus (red) and minus (purple) strand of piRNA cluster #4. Whiskers represent the 5–95 percentile. ***: *p* < 0.001; Kruskal-Wallis test. (**b**) Top right: The percentage of reads of a specified length mapping to the corresponding piRNA clusters. (**b**) Bottom: Sequence logo representation based on entropy for reads mapping to representative piRNA clusters: #1, #2, and #3 shows strong 1T bias. Dark grey: Cluster #4. (**c**) Sequence Logo representation of antisense (top) and sense (bottom) mapping reads on cluster #4. (**d**) Ping-pong signature (overlap of exactly 10 nucleotides counting from the 5′ side for piRNAs mapping opposite strands from a same cluster). (**e**) Genome browser representation of sequencing reads mapping piRNA cluster #4 (region). Reads corresponding to the minus and plus strands are represented in purple and red, respectively. (**f**) Detailed view of a small region consistent with phased piRNA production from a minus-strand derived transcript, and responder piRNAs produced from antisense transcripts by a presumable ping-pong amplification cycle.

**Figure 3 ncrna-05-00019-f003:**
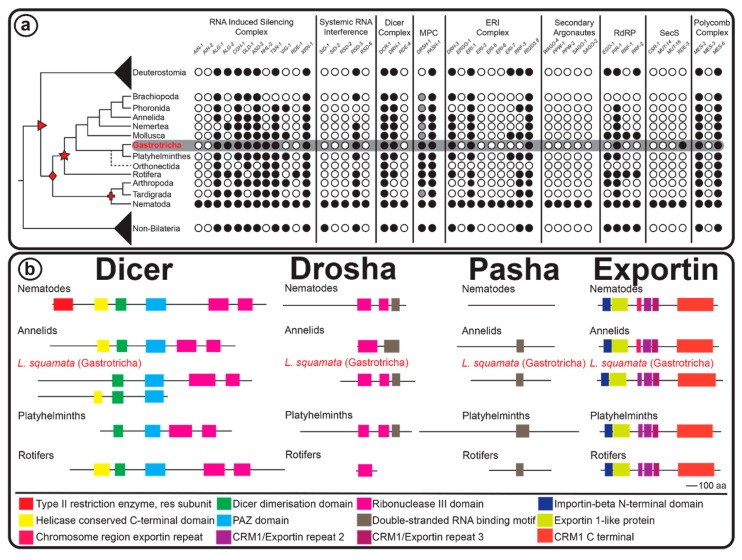
miRNA & piRNA biogenesis proteins and RNAi machinery. (**a**) distribution of RNAi proteins across animals. The cladogram to the left indicates the currently accepted phylogenetic relationships among animal taxa [20,21,54,55,56], and drawings on particular nodes are according to Figure 1d. Black circles denote the presence of proteins from different small RNA complexes, while white circles illustrates protein absence, and grey circles represent a deduced, but not actually detected presence of Drosha based on the presence of bona fide miRNAs. A protein is categorized as present whether we could find it in at least one species from each phylogenetic group. See Appendix A for detailed description of presence/absence of RNAi proteins). Gene names at the top are based on *C. elegans* gene nomenclature. Abbreviations: MPC = Microprocessor Complex; RdRP = RNA dependent RNA polymerases; and SecS = Secondary RNAi Silencing. (**b**) Domain composition of miRNA proteins among selected spiralian and nematode species. The domain composition of Dicer, Drosha, Pasha, and Exportin discovered by reciprocal Blast searches were annotated using Pfam database [57]. All proteins are from genome-derived predicted proteins, except for *L. squamata*. Hence, some proteins might be not full-length due to incomplete assembly.

**Figure 4 ncrna-05-00019-f004:**
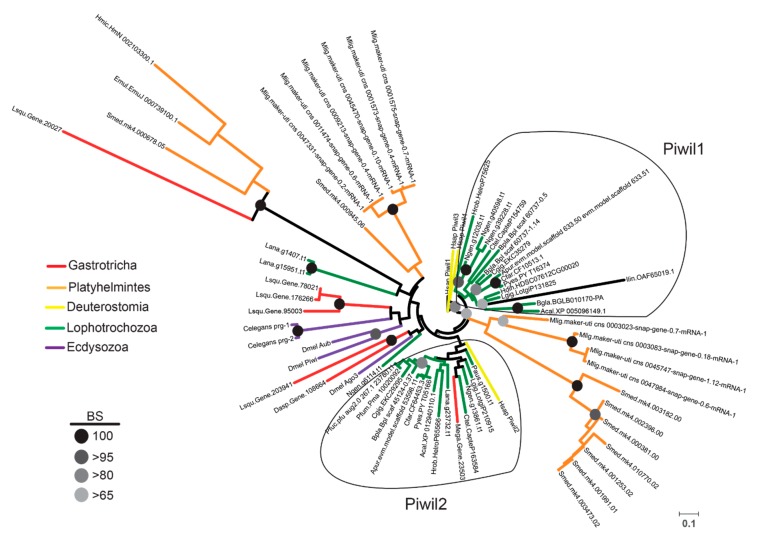
Phylogenetic analysis of PIWI-like proteins. Maximum likelihood analysis of protostome PIWI-like proteins with gastrotrich counterparts highlighted in red. Bootstrap values of particular clades are represented as black-to-grey circles as shown on the lower left. The scale bar on the lower right indicates amino acid substitution rate per site. Color coding on tree branches is shown as follows: Red = gastrotrichs; orange = platyhelminths; yellow = deuterostomes, specifically human PIWI proteins; green = lophotrochozoans (comprising annelids, mollusks, nemerteans, brachiopods, and phoronids); and violet = ecdysozoans (*C. elegans* and *D. melanogaster*).

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
