# Peer review of "Evolutionary Implications of the microRNA- and piRNA Complement of Lepidodermella squamata (Gastrotricha)"

_ncrna, 2019, doi:10.3390/ncrna5010019_

Round 1

Reviewer 1 Report

This work describes the microRNA and piRNA set of Lepidodermalla (Gastroticha). These studies are important in RNA evolutionary biology as it help us to understand the evolutionary origin of the RNA interference machine (in its different flavours). For that reason, I believe this study should be published. On the other hand I have some concerns regarding the annotation procedure and other technical issues that I hope the authors take into account.

MAJOR COMMENTS

The authors make an interesting effort to avoid referring to miRBase. First of all, I have no relationship with miRBase, but whether we like it or not, it is still the standard repository of microRNA sequences. Among other things, miRBase follows the nomenclature agreed by Victor Ambros and David Bartel among others (PMID: 12592000) and that it is widely used by the microRNA community. Also, miRBase has an external review committee, and it is part of RNAcentral, not to mention that major annotation consortia and even companies rely on their annotations. Although I believe that MirGeneDB is a very useful resource and that the curated lists are of great scientific value and rigour, I also believe that an alternative annotation system will lead to several problem, including a naming inconsistency across different databases.

Additionally, I notice that at least one of the authors is behind MirGeneDB itself which creates a secondary problem: the annotation and validation of microRNAs is done by the same authors that propose the sequences. In my opinion, sequences should be annotated and indexed in miRBase, and then a high-confident well-annotated set may be provided in MirGeneDB. As I said, I do think MirGeneDB is useful and I actually use it and cite it. But I don’t think MirGeneDB is an independent and internationally supported alternative to the well established miRBase database.

Related with the previous points, the authors say that the microRNA complement will be available in ‘one of the next releases’ of MirGeneDB. However, as a reviewer I would like to have a look at them. Is this data available somewhere for peer-review?

The discovery/annotation pipeline used was miRCandRef, which I personally didn’t know. When annotating microRNAs in a not particularly well annotated organisms may be useful to use alternative methods in addition. How about miRDeep? I notice one of the authors is the developer of this pipeline and it should be pretty straightforward to them to implement this algorithm.

From the data provided in Figure 1a one can not conclusively determine that Gastroticha are closer to flatworms than to nematodes. The number of differences/matches is small and the evidence can only be minor. As a matter of fact, the 4 microRNAs present in C. elegans and absent in Gastroticha and the flatworms are nematode specific (as reported in MirGeneDB), and therefore they store no taxonomic information. The specific microRNA between Gastroticha and flatworms (mir-1992) may have been secondarily lost. The authors refer to a previous work showing that secondary lost is rare. However, it is not impossible, and one single event cannot be used to define an entire phylogenetic relationship. I’m not saying the authors are not right, I’m just pointing out that the paper should reflect the level of certainty around their claims. I suggest that the authors give a more open message like ‘suggest the evidence’ rather than ‘supports a close relationship’ across the text.

In Figure 1c, the let-7 alignment reveals that the Lepidodermella sequence is quite divergent. Are the authors sure that this is the bona fide let-7? Perhaps a genomics context analysis may reveal further evidence.

Last, but not least, I found out that the author’s contributions may reveal a case of guest authorship. In the journals I’m an editor this is not acceptable. However, as I am not aware of the full editorial policy of this journal I will leave it to the editors to decide on it. More specifically, three of the authors did not contribute intellectually to the development of the work or the writing up.

MINOR COMMENTS:

Line 76: ‘the utility of miRNA as phylogenetic marker’ should be ‘the utility of miRNA as phylogenetic markers’

Figure 4 tree formatting is probably not optimal. A standard (not circular) rooted tree would make text (and clustering) more visible and informative. Also, bootstrap values should be placed in numbers in the branches.

p { margin-bottom: 0.25cm; line-height: 120%; }a:link { }

Author Response

Thank you very much for the review. Find attached detailed answers (in BOLD)

Reviewer 2 Report

In the current manuscript “Evolutionary implications of the first microRNA- and piRNA complement of Lepidodermella squamata (Gastrotricha)” authors analyzed the small RNA sequence data and RNAi protein machinery of L. squamata to support the Gastrotrichs phylogenetic position among nematodes and flatworms. The Small RNAs sequences and their mechanism are among the fast evolving systems, adjusting according to the morphological complexity of a given organism. Use of such a system (small RNA machinery) to study the evolution of a species is highly advantageous when conventional methods such as morphological phylogeny are not successful. I am convinced that the current study has well demonstrated this by showing that based on miRNAs, piRNAs and RNAi protein machinery the L. squamata, a Gastrotricha, has a close relationship to flatworms and not to nematodes.

There are few changes that should be addressed before the manuscript is accepted (see below).

Minor corrections:

Comment 1: (Line 30) Please provide the full form of RNAi (RNA interference)

Comment 2: (Line 96) Please remove the (t)

“complement (miRNA and piRNA) as well as t the corresponding protein machinery”

Comment 3: (Line 105) Please provide SRA Accession number

Comment 4: (Line 180) Please correct the format and remove the repeated word

“was found in L. squamata thatis is distinct”

Comment 5: (Line 215) Please remove the (t)

“but not to conclude t that a”

Comment 6: (Line 223) Please remove the (W)

“proteins w in flatworms”

Comment 7: (Line 225) Please correct the word (ws)

“cytoplasm in C. elegans [55], ws also more”

Comment 8: (Line 329) information is missing in the sentence

“of them - at some extend - with similarity”

Comment 9: (Line 409) provide the details of Pfam database version or release date used for the analysis.

Major corrections:

Comment 10: line 27 in abstract: Small non-coding RNA data on e.g. microRNAs (miRNAs) and 26 PIWI-interacting RNAs (piRNA) may help to resolve this long-standing question.

I too agree with the above statement, however, authors may consider to rewrite this sentence in a more subtle way. I think that “resolving this long-stating question” is a strong statement, instead, authors can use “miRNAs and piRNAs sequence homology may offer additional support the to the phylogenomic analyses

Comment 11: Materials and Methods section line 393, please provide the details of the NCBI SRA database.

Comment 12: (line 400 – 402) in order to carry out blast search for RNAi proteins across metazoan species: why did authors consider only C. elegans? Even though the RNAi pathway is well studied in C. elegans, the base of the current analysis is to address the relationship or phyletic position of Gastrotrichs against the nematodes and flatworms. I would suggest to the authors to consider carrying out the reciprocal Blast search against a list of RNAi proteins retrieved from a flatworm species such as Planaria.

Comment 13: section “4.4 Identification and domain architecture of RNAi proteins” Hence the NCBI databases and BLASTp are frequently updated, please provide the release date and version respectively.  

Comment 14: Figure 1a, b & c, Figure 2, Figure 3a: Please increase the font size in figures.

Comment 15: (line 426-431) the information about Supplementary figure S1-S4 is missing in Supplementary Material 

Author Response

(The authors gave the same response as above.)

Round 2

Reviewer 1 Report

I thank the authors for taking into account my comments. I think most points are now clarified and that the manuscript is of publication quality. Yet, I still have some minor points/comments to the authors:

1) I appreciate the concerns the authors have about the quality of the annotation of miRBase sequences. I do agree. And I also agree with the authors that "the role of miRBase is to serve as an open access repository for published miRNA sequences". That is precisely why sequences should be sent to miRBase to have a centralized, unique and reliable database to store microRNAs described by the community including false negatives, wrong sequences and any previous record. That what makes a database valuable. There are numerous erroneous records in GeneBank yet we don't need alternative catalogs with alternative nomenclatures. But as I said, curated lists like that of MirGeneDB are handy. Also, when publishing papers one can give provisional names and send them off to miRBase which, in turn, will update miRBase and provide 'official' names to the authors. That's what happened in my previous publications. If, for instance, I discover a new microRNA and I call it mir-6666 just because, it may be that this name has being assigned to another microRNA and may eventually appear in miRBase (and subsequently in ENSEMBL, UCSC and NCBI) and create a case of two different sequences with the same name. I hope the authors understand my concerns here.

2) The authors toned down one sentence of the manuscirpt as :

The comparison of miRNAs, piRNAs and RNAi protein machinery, to those of flatworms and nematodes together supports a close relationship of gastrotrichs to flatworms and not to nematodes.

p { margin-bottom: 0.25cm; line-height: 115%; }a:link { }

In my opinion this is not significantly different to what they had before. I suggest something in the line of:

The comparison of miRNAs, piRNAs and RNAi protein machinery, to those of flatworms and nematodes together is compatible with a closer relationship of gastrotrichs to flatworms rather than to nematodes.

Of course, the decision is entirely to the authors. My point here is that, if gastrotichs and flatworms are definitively found to be closely related, the authors will be able to claim some priority in that finding. But if the opposite pattern is found, the authors will be safe as they based their findings on the evidence available. We all remember cases (not to mention here) of careers heavily affected by strong claims based on weak evidence, particularly in the evolutionary relationships between animals. But again, this is entirely the decision of the authors. Just wanted to share my thoughts.

3) I am sure that the authors contributed significantly to the work and should be listed as authors. However, the current section on author's contributions does not reflect that. The authors can read from the MDPI webpage that:

Each author is expected to have made substantial contributions to the conception or design of the work; acquisition, analysis, or interpretation of data; the creation of new software used in the work; and/or writing or substantively revising the manuscript. In addition, all authors must have approved the submitted version (and any substantially modified version that involves the author’s contribution to the study); AND agrees to be personally accountable for the author’s own contributions and for ensuring that questions related to the accuracy or integrity of any part of the work, even those in which the author was not personally involved, are appropriately investigated, resolved, and documented in the literature. Note that acquisition of funding, collection of data, or general supervision of the research group do not, by themselves, justify authorship.Those who contributed to the work but do not qualify for authorship should be listed in the acknowledgements.

p { margin-bottom: 0.25cm; line-height: 115%; }a:link { }

This is not a minor point and the authors should clarify authors' contributions in the manuscript.

Author Response

Reviewer 1

I thank the authors for taking into account my comments. I think most points are now clarified and that the manuscript is of publication quality. Yet, I still have some minor points/comments to the authors:

1) I appreciate the concerns the authors have about the quality of the annotation of miRBase sequences. I do agree. And I also agree with the authors that "the role of miRBase is to serve as an open access repository for published miRNA sequences". That is precisely why sequences should be sent to miRBase to have a centralized, unique and reliable database to store microRNAs described by the community including false negatives, wrong sequences and any previous record. That what makes a database valuable. There are numerous erroneous records in GeneBank yet we don't need alternative catalogs with alternative nomenclatures. But as I said, curated lists like that of MirGeneDB are handy. Also, when publishing papers one can give provisional names and send them off to miRBase which, in turn, will update miRBase and provide 'official' names to the authors. That's what happened in my previous publications. If, for instance, I discover a new microRNA and I call it mir-6666 just because, it may be that this name has being assigned to another microRNA and may eventually appear in miRBase (and subsequently in ENSEMBL, UCSC and NCBI) and create a case of two different sequences with the same name. I hope the authors understand my concerns here.

Answer: Thank you for this comment. We have already in the previous round of reviews acknowledged this point and submitted our complement to miRBase. As for now we are waiting for a reply and have - in the meantime - named novel miRNAs with “NOVEL”, provisional names.

2) The authors toned down one sentence of the manuscirpt as :

“The comparison of miRNAs, piRNAs and RNAi protein machinery, to those of flatworms and nematodes together supports a close relationship of gastrotrichs to flatworms and not to nematodes.“

In my opinion this is not significantly different to what they had before. I suggest something in the line of:

“The comparison of miRNAs, piRNAs and RNAi protein machinery, to those of flatworms and nematodes together is compatible with a closer relationship of gastrotrichs to flatworms rather than to nematodes.“

Of course, the decision is entirely to the authors. My point here is that, if gastrotichs and flatworms are definitively found to be closely related, the authors will be able to claim some priority in that finding. But if the opposite pattern is found, the authors will be safe as they based their findings on the evidence available. We all remember cases (not to mention here) of careers heavily affected by strong claims based on weak evidence, particularly in the evolutionary relationships between animals. But again, this is entirely the decision of the authors. Just wanted to share my thoughts.

Answer: Thank you for this point. We are confident that the data we analyzed supports the close relationship of gastrotrichs to flatworms.

3) I am sure that the authors contributed significantly to the work and should be listed as authors. However, the current section on author's contributions does not reflect that. The authors can read from the MDPI webpage that:

“Each author is expected to have made substantial contributions to the conception or design of the work; acquisition, analysis, or interpretation of data; the creation of new software used in the work; and/or writing or substantively revising the manuscript. In addition, all authors must have approved the submitted version (and any substantially modified version that involves the author’s contribution to the study); AND agrees to be personally accountable for the author’s own contributions and for ensuring that questions related to the accuracy or integrity of any part of the work, even those in which the author was not personally involved, are appropriately investigated, resolved, and documented in the literature. Note that acquisition of funding, collection of data, or general supervision of the research group do not, by themselves, justify authorship.Those who contributed to the work but do not qualify for authorship should be listed in the acknowledgements.“

This is not a minor point and the authors should clarify authors' contributions in the manuscript.

Answer: Thank you very much for making this valid point. We have now expanded the authors’ distribution (beyond the previously strictly followed scheme) significantly to:

“Author Contributions: Conceptualization, B.F.,M.R.F., L.B., A.H. ; Methodology, B.F., J.P.T, F.A.; Software, B.F., J.P.T, F.A.; Sample & Data Acquisition, B.F., A.H, L.B. Data Curation, B.F., J.P.T., F.A.; Writing-Original Draft Preparation, B.F., J.P.T., F.A.; Writing-Review & Editing, all authors. Visualization, B.F., J.P.T., F.A.; Project Administration, B.F.; Funding Acquisition, M.R.F., L.B. and A.H.”

Reviewer 2 Report

My previously minor comments have been completely addressed.  I consider that the quality and impact of the work presented now deserve publication. Congratulations on a nice piece of work.

Author Response

Thank you very much for your constructive review.

Non-Coding RNA EISSN 2311-553X Published by MDPI AG, Basel, Switzerland RSS E-Mail Table of Contents Alert
Back to Top